# Molecular diversity and genetic structure of *Saccharum* complex accessions

**Carolina Medeiros, Thiago Willian Almeida Balsalobre, Monalisa Sampaio Carneiro**<sub></sub>*

Departamento de Biotecnologia e Produção Vegetal e Animal, Centro de Ciências Agrárias, Universidade Federal de São Carlos, Araras, São Paulo, Brasil

* monalisa@ufscar.br

**Data Availability Statement:** All relevant data are within the manuscript and its Supporting Information files.

**Funding:** This work was supported by grants from the FINEP (Financiadora de Estudos e Projetos),

## Abstract

Sugarcane is an important crop for food and energy security, providing sucrose and bioethanol from sugar content and bioelectricity from lignocellulosic bagasse. In order to evaluate the diversity and genetic structure of the Brazilian Panel of Sugarcane Genotypes (BPSG), a core collection composed by 254 accessions of the *Saccharum* complex, eight TRAP markers anchored in sucrose and lignin metabolism genes were evaluated. A total of 584 polymorphic fragments were identified and used to investigate the genetic structure of BPSG through analysis of molecular variance (AMOVA), principal components analysis (PCA), a Bayesian method using STRUCTURE software, genetic dissimilarity and phylogenetic tree. AMOVA showed a moderate genetic differentiation between ancestors and improved accessions, 0.14, and the molecular variance was higher within populations than among populations, with values of 86%, 95% and 97% when constrasting improved with ancestors, foreign with ancestors and improved with foreign, respectively. The PCA approach suggests clustering in according with evolutionary and Brazilian breeding sugarcane history, since improved accessions from older generations were positioned closer to ancestors than improved accessions from recent generations. This result was also confirmed by STRUCTURE analysis and phylogenetic tree. The Bayesian method was able to separate ancestors of the improved accessions while the phylogenetic tree showed clusters considering the family relatedness within three major clades; the first being composed mainly by ancestors and the other two mainly by improved accessions. This work can contribute to better management of the crosses considering functional regions of the sugarcane genome.

## Introduction

Sugarcane, a high efficiency photosynthetic grass, is important for economy of many countries in the tropics and subtropics, playing a central role as a primary sugar-producing crop and has major potential as a bioenergy crop [1–3]. The modern sugarcane cultivars originate from the *Saccharum* complex, which gathers two wild *Saccharum* species (*S. spontaneum* and *S. robustusm*), four cultivated species (*S. officinarum*, *S. sinense*, *S. barberi* and *S. edule*) and four related interbreeding genera (*Erianthus*, *Miscanthus*, *Narenga* and *Sclerostachya*) [4–7]. The

FAPESP (Fundação de Amparo à Pesquisa de São Paulo, 08/57908-6) and CNPq, Conselho Nacional de Desenvolvimento Científico e Tecnológico, 574002/2008-1). CM received a master's fellowship from CAPES (Coordenação de Aperfeiçoamento de Pessoal de Nível Superior – Finance Code 001).

**Competing interests:** The authors have declared that no competing interests exist.

*Saccharum* species present large genome and variation in the number of chromosomes [8–10]. This complexity was inherited by modern sugarcane cultivars, which present a variable level of ploidy, frequent aneuploidy, and large genome size around 10 Gb [10–12].

The first interspecific hybridizations occurred among *S. officinarum* and *S. spontaneum* species, followed by successive backcrosses with *S. officinarum* aiming to recover the sucrose genes [4,13]. According to this initial breeding strategy, naturally few accessions were used at the crosses and approximately 80% of the genome of current sugarcane cultivars came from *S. officinarum*, 10–15% from *S. spontaneum* and the remaining 5–10% being recombinant chromosomes [14,15]. Differently of *S. officinarum*, the accessions of *S. spontaneum* present low sugar content, high biomass production and resistance to some diseases [2,16]. Thus, it is an important genetic background to increase biomass production and have been used into plant breeding for energy cane purpose [13,17]. This energy cane with more fiber content and low sugar production could be an efficient source for second-generation ethanol production [18,19]. Furthermore, the higher rates of biomass and/or sucrose production can be obtained through better management of genetic resources present in germplasm banks and core collections.

The pre-breeding strategy to choose parents for crosses is an important step to increase the probability of obtaining more productive cultivars. Although morphological and agronomical characterization plays an important role in the classification and organization of germplasm accessions, errors may occur since vegetative traits are influenced by environmental effects, showing continuous variation and a high degree of plasticity, and which many times do not reflect the real genetic diversity of the *Saccharum* spp. accessions [20]. So, the molecular profile could be used to complement the morphological characterization and identify in a more reliable way better combinations between accessions for crosses according to breeding goals [18,21,22]. Molecular markers are useful tools to detect variations directly in the genome and have been used to investigate the genetic diversity of *Saccharum* spp. accessions [23,24]. However, few studies performed molecular characterization of sugarcane core collections with functional markers, most of them evaluated non-coding or repeating regions of the genome and may not be useful about traits of interest to the breeders [25,26]. Even when functional molecular markers were used, the number of *Saccharum* spp. accessions evaluated was not more than 181 [27]. Clearly, there is a need to expand the characterization of larger and more representative *Saccharum* complex collections with functional markers, bringing together both alleles under bottleneck effect and those that may be new sources of variation for target traits.

TRAP (Target Region Amplification Polymorphism) and EST-SSR (Simple Sequence Repeats from Expressed Sequence Tag) molecular markers, beside those identified through genetic mapping, could be used to screening collections into functional regions of genome [25]. TRAP markers are interesting because they search for polymorphisms around genes that may be under selection process [28,29]. Furthermore, this approach may indicate accessions for crosses according to molecular profile and, consequently, guide introgression of the new variants for traits of interest [25,29]. Sucrose and lignin are target traits to sugarcane and energy cane breeding programs; increase sugar content is one of the main goals of sugarcane breeding programs around the world [30], while decreasing lignin content may facilitate cellulose saccharification for second-generation ethanol production from both sugarcane and energy cane [19,31]. Sucrose and lignin traits are governed by some genes and metabolic pathways described in the literature [16,21,30–32], so the use of TRAP markers based on these genes may be a valuable tool to characterize *Saccharum* spp. accessions and research new allelic variants. Therefore, in this current assignment our objectives were to (i) characterize a core collection of sugarcane composed by 254 accessions of the *Saccharum* complex, and (ii) perform diversity and population structure assessments, using genotyping data obtained through

TRAP markers based on the sucrose and lignin genes. We discuss these results in the context of how functional markers are useful to report evolutionary and breeding history of sugarcane.

## Materials and methods

### Plant material and DNA extraction

In this study, a total of 254 accessions (S1 Table) of the Brazilian Panel of Sugarcane Genotypes (BPSG) were used. BPSG is a mini core collection from germplasm bank of the RIDESA (Interuniversity Network for the Development of Sugarcane Industry) and consisted of 81 ancestors accessions (A) (75 accessions from *Saccharum* spp. and 06 from *Erianthus* spp.), 137 hybrids from Brazilian breeding programs (BB) and 36 hybrids from Foreign breeding programs–Foreign Hybrids (FH) [33]. The BPSG accessions were chosen according to the following criteria: i) relevant Brazilian cultivars, ii) main parents for Brazilian breeding programs; iii) cultivars from countries that grow sugarcane; iv) parents used in mapping programs [34,35]; and v) representatives of the species from which the *Saccharum* complex originated. The genetic variability present into BPSG, for the most part, was a genetic basis for Brazilian sugarcane breeding programs. The stalks of the accessions were collected and total genomic DNA was extracted from a fresh meristem cylinder as proposed by Al-Janabi et al. [36]. DNA concentration was estimated by a Nanodrop One spectrophotometer (Thermo Scientific, Wilmington, DE, USA) and then the samples were stored at −20˚C until further use.

### TRAP markers, genotyping and polymorphism analysis

To compose TRAP markers four arbitrary and five fixed primers were used (S2 Table). The arbitrary primers were adapted of Li and Quiros [37], Alwala et al. [21] and Suman et al. [30]. The three fixed primers associated with sucrose metabolism genes were based on Alwala et al. [21] (*sucrose synthase* (SuSy), *sucrose phosphate synthase* (SuPS) and *starch synthase* (StSy)) while two fixed primers associated with lignin metabolism genes were based on Suman et al. [30] (*caffeic acid O-methyltransferase* (COMT) and *ferulate-5-hydroxylase* (F5H)). Thus, eight TRAP markers were performed based on high percentage of polymorphism showed by the reference studies: StSy + Arbi2, StSy + Arbi3, SuPS + Arbi2, SuPS + Arbi3, SuSy + Arbi1-A, SuSy + Arbi2 for sucrose metabolism and COMT+Arbi1-S and F5H+Arbi1-S for lignin metabolism. The PCR were performed on Mastercycler thermocycler (Eppendorf, Westbury, NY, USA) according to the protocols described by Alwala et al. [21] and Suman et al. [30] for TRAP markers related with sucrose and lignin metabolisms, respectively. After PCR, the amplified products were run on 6.5% (w/v) polyacrylamide denaturing gel for 4.0 h at 65 W and silver staining procedure was employed to detect the fragments as described by Creste et al. [38]. The fragments were scored as "1" for presence and "0" for absence, in all accessions. Only clearly distinguishable fragments were scored. For each TRAP marker, the presence of exclusive fragments was investigated. Through the binary matrix, the PIC (Polymorphism Information Content) and DP (Discriminatory Power) values were calculated according to Botstein et al. [39] and Tessier et al. [40], respectively. PIC was used as a tool to measure the information of a given marker locus for the pool of accessions, while DP was used as a measure of marker efficiency for the purpose of identification of accession, i.e., the probability that two randomly chosen individuals have different patterns [41].

### Sequence annotation

The available sequences that gave rise to fixed primers of TRAP markers were used to annotation (S3 Table). To found homologies the initial sequences from Genbank were blasted against

the NCBI non-redundant database via BLASTX and against the *Sorghum bicolor* database via the Phytozome website [42]. The metabolic pathways and biochemical reactions were also verified through the InterMine repository present on the Phytozome.

### Genetic structure

The genetic structure of BPSG was investigated using different methods: i) analysis of molecular variance (AMOVA); ii) Principal component analysis (PCA); iii) a Bayesian model-based method using STRUCTURE software; and iv) genetic dissimilarity and phylogenetic analysis. AMOVA was performed by the GenAIEx software [43] to quantify the degree of differentiation and distribution of the genetic variability between and within of predefined cases: a) ancestors accessions (A group) and accessions from Foreign breeding programs (FH group); b) ancestors accessions (A group) and accessions from Brazilian breeding programs (BB group); and c) accessions from Foreign breeding programs (FH group) and accessions from Brazilian breeding programs (BB group). PCA was performed in the R software [44] through the FactoMineR [45] and factoextra [46] packages and their respective functions PCA and fviz_pca_ind using raw data from genotyping of TRAP markers. The analysis with STRUCTURE software [47,48], to verify the number of subpopulations (*k*) and the membership proportion (*Q*), was performed considering the 248 accessions of the *Saccharum* genus of BPSG, i.e. without accessions of *Erianthus* genus. The *k* was set from 1 to 10 (*k*-value), with 10 iterations at a 100,000 burning period and 200,000 Markov Chain Monte Carlo (MCMC) repeats. The STRUCTURE HARVESTER software was used to find the best values of *k* and *Δk* [49]. Finally, the pair-wise dissimilarity among the accessions of the *Saccharum* genus was performed in the R software according to the Jaccard coefficient (Dissimilarity = 1 –Similarity) and a phylogenetic tree was build according to the neighbor-joining (NJ) method with 1,000 bootstrapping through ggtree and ape packages [50, 51]. To verify if the number of TRAP fragments used to estimate the genetic dissimilarities between accessions was adequate in terms of accuracy, the bootstrap resample technique [52] was applied as in Manechini et al. [23]. Briefly, an exponential function was adjusted to estimate the number of markers needed to assure that the CV associated with the dissimilarity estimates were lesser or equal to 10%, a threshold considered acceptable in this research. The median of the coefficient of variation estimates were used to evaluate the accuracy of the dissimilarity values [53].

## Results

### TRAP markers polymorphism and population differentiation

The results regarding the total number of fragments, number of polymorphic fragments, percentage of polymorphism, PIC and PD values for each of the eight TRAP markers used in this study are summarized in S4 Table. A total of 595 fragments were obtained of which 584 were polymorphic. The number of fragments for each TRAP markers ranged from 44 (SuPS + Arbi2) to 88 (SuSy + Arbi1-A) with an average of 74.37 fragments per locus. The polymorphism percentage was high (> 90%), ranging from 94.64% (SuPS + Arbi3) to 100% (SuPS + Arbi2, COMT + Arbi1-S and F5H + Arbi1-S). The averages of PIC and PD values were 0.97 and 0.98, respectively.

Putative exclusive TRAP fragments were observed for A and BB predefined groups and represented 11.64% of the total polymorphic fragments (S5 Table). The SuSy + Arbi2 showed the largest number of putative exclusive fragments (18), all present in the representative accessions of *Erianthus* spp. This specie was the one that had more putative exclusive fragments (49), followed by *S. spontaneum* (08), *S. robustum* (06), *S. officinarum* (01) and *S. barberi* (01). In the

**Table 1. Analysis of molecular variance (AMOVA) between predefined groups A, BB and FH of the Brazilian Panel of Sugarcane Genotypes (BPSG).**

|  | Source of variation | d.f. | Sum of squares | Variance components | Porcentage of variation |
|---|---|---|---|---|---|
| BB and A | Among population | 1 | 944.93 | 9.28 | 14% |
|  | Within populations | 209 | 12345.46 | 59.07 | 86% |
|  | Total | 210 | 13290.39 | 68.35 |  |
|  |  |  |  |  | Genetic differentiation ($\Phi_{PT}$): 0.14* |
| FH and A | Among population | 1 | 242.44 | 3.38 | 5% |
|  | Within populations | 113 | 7943.11 | 70.29 | 95% |
|  | Total | 114 | 8185.55 | 73.68 |  |
|  |  |  |  |  | Genetic differentiation ($\Phi_{PT}$): 0.05* |
| BB and FH | Among population | 1 | 138.34 | 1.48 | 3% |
|  | Within populations | 174 | 8761.99 | 50.36 | 97% |
|  | Total | 175 | 8900.33 | 51.83 |  |
|  |  |  |  |  | Genetic differentiation ($\Phi_{PT}$): 0.03* |

d.f.: degrees of freedom.

*$P < 0.001$.

BB group, three putative exclusive fragments were present, representing 0.51% of the total polymorphic fragments.

Considering all predefined groups (A, BB and FH), the AMOVA results revealed that the molecular variance found by TRAP markers was higher within populations than among populations (Table 1). The genetic differentiation value ($\Phi_{PT}$) obtained between A and BB groups was 0.14, which means that 14% of the total variation found by the TRAP markers was distributed between these two groups, while 86% was within them. The $\Phi_{PT}$ values obtained between A and FH groups ($\Phi_{PT} = 0.05$) and between BB and FH groups ($\Phi_{PT} = 0.03$) were lower than that observed between A and BB groups. In addition, $\Phi_{PT}$ values were significant for all comparisons between groups ($P < 0.001$).

## Principal component analysis

Principal component analysis (PCA) was firstly performed based on 595 TRAP fragments with all 254 accessions of BPSG, which includes accessions of predefined groups A, BB and FH (Fig 1A). Considering that the panel under study presents accessions of two genera, *Saccharum* and *Erianthus*, a second PCA was performed without accessions of the genus *Erianthus* (using 546 TRAP fragments) aiming to detect some clustering among the *Saccharum* accessions (Fig 1B).

Thereby, in the Fig 1A the first two principal components, PC1 and PC2, explained 17.8% of the total variability expressed among accessions. According to PC1 it is possible to note that *Erianthus* accessions (75//09 ERIANTHUS, H. KAWANDANG, IM76-227, IN84-73, IN84-77 and IN84-83) were grouped in an isolated cluster from the others accessions. In addition, *S. spontaneum* accessions were allocated together (IN84-58, IN84-82, IN84-88, KRAKATAU and SES205A). In contrast, accessions of the FH group were distributed in non-clustered way; some FH accessions were allocated near to accessions of the A group (for example, CR72/106 and US60-31-3), while others were closer to accessions of BB group (for example, NCo-310 and EK28). The BB group showed a tendency of clustered greater than A and FH groups, and it is possible to note two subgroups within the group.

Already in the second PCA, PC1 and PC2 explained 12.7% of the total variability expressed among accessions (Fig 1B). The accessions of A group were distributed over PC1, being some

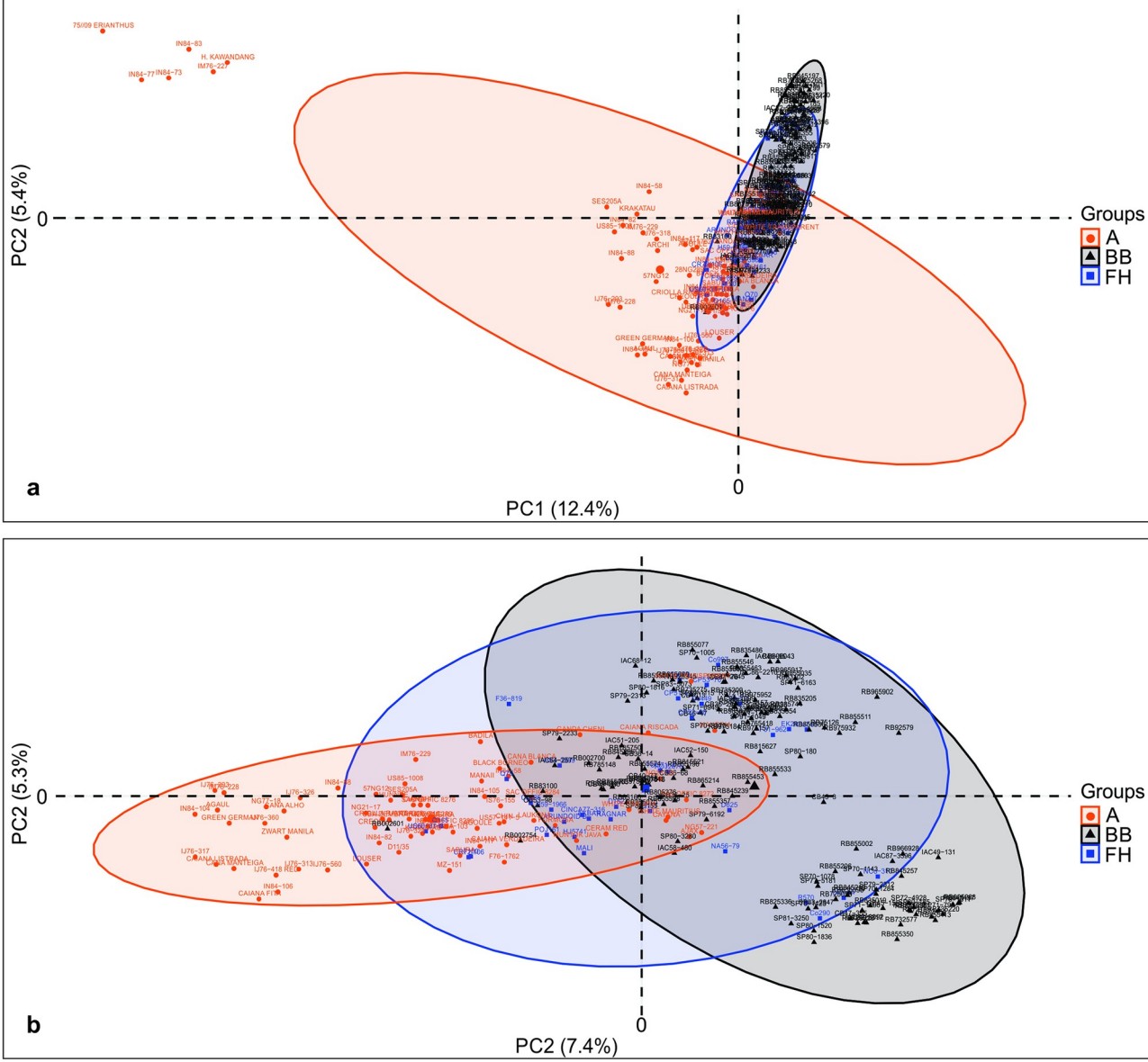

**Fig 1. Principal Components Analysis (PCA) of the Brazilian Panel of Sugarcane Genotypes (BPSG) based on TRAP markers.** (a) PCA performed with all 254 accessions of the BPSG. (b) PCA performed with 248 accessions of the BPSG, i.e., without accessions representatives of the genus *Erianthus*. The different colors indicates the predefined groups: ancestors accessions (A) in orange; accessions of *Saccharum* spp. hybrids from Brazilian breeding programs (BB) in black; accessions of *Saccharum* spp. hybrids from foreign breeding programs (FH) in blue. The A group in the Fig 1A was composite by ancestors accessions of the genus *Saccharum* and *Erianthus*, while in the Fig 1B, the A group was composite only by accessions of the genus *Saccharum*.

accessions of *S. officinarum* (for example, WHITE TRANSPARENT, CAIANA RISCADA, SAC OFFIC 8272, NG21-21, NG57-221, CAYANA, WHITE MAURITIUS and AJAX) closely positioned with accessions originated from breeding programs. In addition, accessions representatives of *S. barberi* (GANDACHENI and WHITE PARARIA) and *S. sinense* (MANERIA) were also nearby of improved accessions. The accessions of FH group were distributed almost equally along PC1 and PC2, which can be observed by the blue ellipse with center near the 0–0

coordinate and also by presence of FH accessions in the four quadrants of the graph. The BB group apparently showed the division of their accessions into two clusters, one with most accessions in the second quadrant and other in the fourth quadrant of the graph. In general, this separation agrees with pedigree information, for example, the RB965917 and RB965902 accessions are full-sibs originated from cross between RB855453 and RB855536, all of them were positioned into cluster at second quadrant. Furthermore, RB845197, RB845210, RB845257, RB855036, RB855002 and RB855113 are full-sibs originated from cross between RB72454 and SP70-1143, all of them allocated into cluster at fourth quadrant. The presence of half-sibs should also contribute to this separation, for example, RB806043, RB815521, RB83102, RB855533, SP71-6163, SP716949, SP81-1763, RB815627, RB815690 and RB835054 accessions sharing the parent NA56-79 and all allocated into cluster at second quadrant. Like- wise IAC87-3396, SP83-2847, RB845197, RB845210, RB845257, RB855036, RB855002, RB855070, RB855113, RB855595 and RB855598 accessions sharing the parent SP70-1143 and all were positioned into cluster at fourth quadrant.

## Structure analysis

According to the STRUCTURE analysis (without accessions of genus *Erianthus*), the best *k* value was two (*Δk* = 399.43, S1 Fig), suggesting that the 248 accessions of genus *Saccharum* could be divided into two subpopulations, P1 and P2, containing 178 and 70 accessions, respectively (Fig 2). P1 had 164 accessions belonging to BB and FH groups and only 14 acces- sions belonging to A group. The ancestors accessions into P1 were representatives of *S. offici- narum* (AJAX, BLACK BORNEO, CAIANA RISCADA, CAYANA, CERAM RED, FORMOSA, LAUKONA, NG21-21, NG57-221, SAC OFFIC 8272, WHITE MAURITIUS and WHITE TRANSPARENT), *S. barberi* (GANDACHENI) and *S. sinense* (MANERIA). In con- trast, P2 had 61 accessions belonging to A group and only nine accessions were improved accessions (AROUNDOID B, CR72/106, Q165, RB83100, RB002601 and US60-31-3, Co285, F150, HJ5741). Therefore, P1 had most of the accessions of BB and FH groups, while P2 had most of accessions of A group. Furthermore, 20 accessions showed probabilities to be part of both subpopulations (Fig 2). Among these, seven accessions were more likely to be allocated in P1 (RAGNAR, BLACK BORNEO, FORMOSA, LAUKONA, POJ161, Q70 and RB002754) and the other 13 accessions were more likely to be included in P2 (ARUNDOID B, BADILA, CAI- ANA VERDADEIRA, CANA BLANCA, Co285, F150, HJ5741, IS76-155, IN84-105, MANAII, Q165, RB83100 and SAC OFFIC 8284).

## Genetic dissimilarity and phylogenetic analysis

The number of TRAP fragments used in this study was sufficient to estimate the pair-wise genetic dissimilarity with an acceptable level of accuracy. Considering the 546 fragments used in this analysis the CV was 8.64% (S2 Fig), under the threshold previously established of 10%. An amount around 400 fragments would be sufficient to obtain a CV average estimate around 10%.

The higher dissimilarity value was found between SES205A (*S. spontaneum*) and CAIANA FITA (*S. officinarum*) accessions (0.62), and the lower dissimilarity value was between CB40- 13 and RB721012 accessions (0.10), both belonging to BB group. The average dissimilarity val- ues within the A, BB and FH groups were 0.36, 0.25 and 0.29, respectively. Considering a sub- division of BB group according to different Brazilian breeding programs, the average dissimilarities were 0.23, 0.24, 0.26 and 0.26 within CB, IAC, RB and IAC subgroups, respec- tively. The highest average dissimilarities were found when A group was compared with FH group (0.34) and BB group (average of 0.34). On the other hand, smaller average dissimilarities

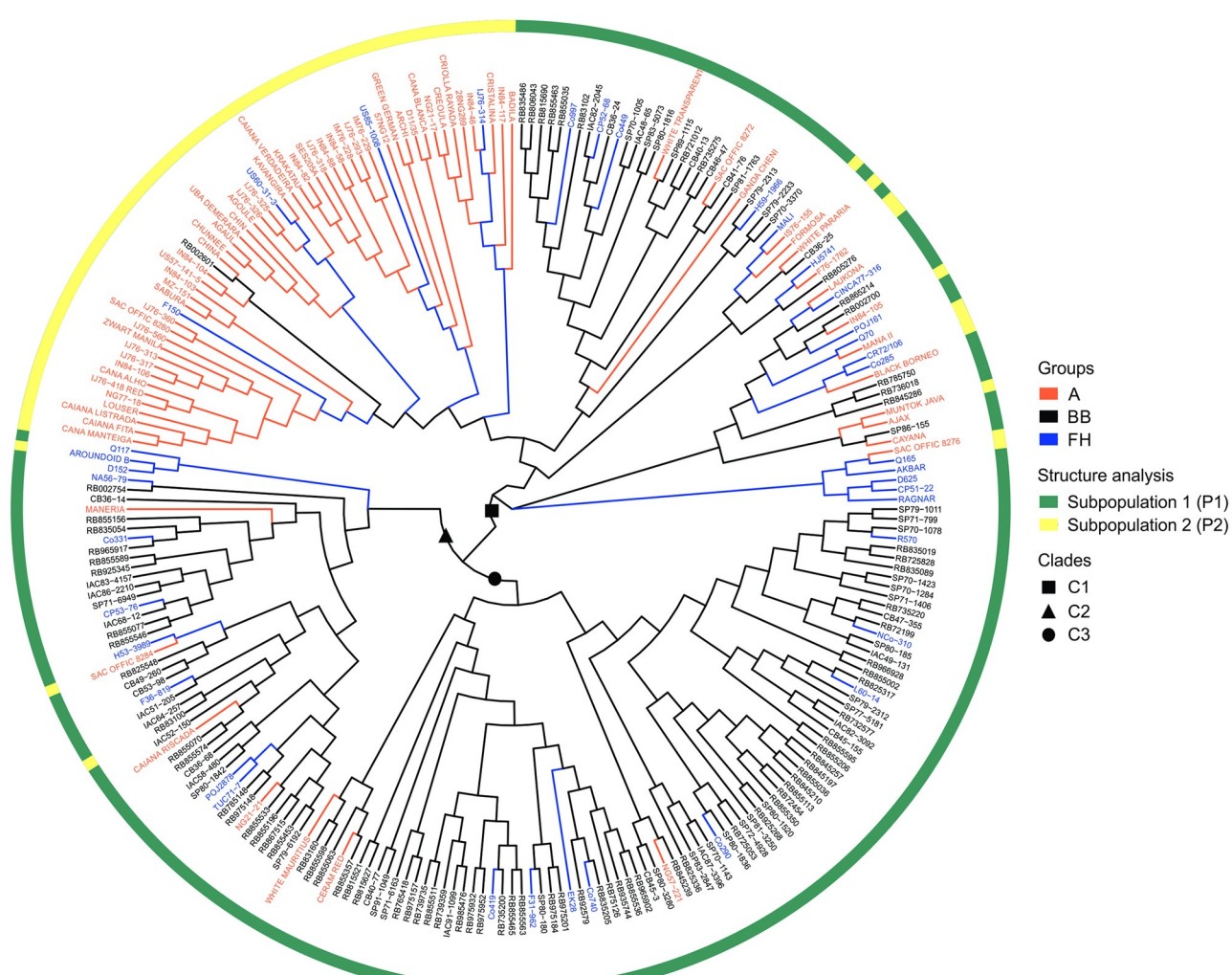

**Fig 2. Phylogenetic tree estimated through neighbor-joining method for 248 accessions of the Brazilian Panel of Sugarcane Genotypes (BPSG).** The names of the accessions belonging to predefined groups were write with different colors: ancestors accessions (A) in orange; accessions of *Saccharum* spp. hybrids from Brazilian breeding programs (BB) in black; and accessions of *Saccharum* spp. hybrids from foreign breeding programs (FH) in blue. The A group was composite by ancestors accessions of the genus *Saccharum*. The circumference around the phylogenetic tree represents the two subpopulations estimated by the STRUCTURE analysis and the green and yellow colors indicate accessions of the P1 and P2 subpopulations, respectively. The three major clades C1, C2 and C3 were indicated within the phylogenetic tree with square, triangle and circle in black.

occurred between and within of the FH group and BB subgroups (CB, IAC, RB and SP), ranged from 0.23 (within CB subgroup) to 0.29 (within FH group and between FH group and SP subgroup).

The phylogenetic tree carried out with accessions of genus *Saccharum* suggests the presence of three major clades (Fig 2). The clade C1 was composed mainly by accessions of A group (68 accessions), followed by 31 BB accessions and 18 FH accessions (AKBAR, CINCA77-316, Co285, Co997, Co449, CP51-22, CP52-68, CR72/106, D625, F150, H59-1966, HJ5741, MALI, POJ161, Q70, Q165, RAGNAR and US60-31-3). On the other hand, the clades C2 and C3 were composed largely for BB accessions. Clade C2 was composed by 32 BB accessions, 10 FH accessions and four A accessions (CAIANA RISCADA, MANERIA, NG21-21 and SAC

OFFIC 8284). Finally, clade C3 had 74 BB accessions, eight FH accessions and three accessions from the A group (CERAM RED, NG57-221 and WHITE MAURITIUS).

Furthermore, the clustering of the phylogenetic tree was similar to the arrangement of accessions in the PCA (Fig 1B) and, in general, the composition of clades was also in agreement with pedigree information. Evidence of this is that some accessions that were widely used as parents by Brazilian breeding programs were grouped at same clade with their progenies of full-sibs or half-sibs. In clade C2, there are three interesting cases: a) F36-819, IAC58-480 and IAC51-205 are half-sibs with the parent POJ2878 in common; b) RB835054, RB83100, RB855533 and SP71-6949 are half-sibs with the parent NA56-79 in common; and c) RB855156, RB855196, RB855070, RB855077, RB855574, RB855589 and RB855453 are half-sibs with the parent TUC71-7 in common. In clade C3, for example, the full-sibs RB845197, RB845210, RB845239, RB845257, RB855036, RB855002, RB855113 and RB855536 were grouped together with their parents SP70-1143 and RB72454. On the other hand, although present to a lesser extent, we also noticed that some full-sib accessions were allocated in different clades and so partially diverging from pedigree information, as in the case of CB40-13 and CB40-77 accessions, being the first positioned in clade C1 and the second in clade C3. The parents POJ2878 and Co290 were positioned in clades C2 and C3, respectively.

## Discussion

TRAP markers have been used in assessment of genetic diversity in plants with complex genomes such as sugarcane and wheat [21,25,29,30,54–60]. The BPSG was composed by accessions representatives of different species of the *Saccharum* complex and also by different hybrids from Brazilian and foreign breeding programs, which constitutes a broad genetic background and allelic pool to be explored and a great breeding value towards the sugarcane genetic improvement. So, the high variability and genome complexity into BPSG contributed to the large number of polymorphic fragments for each TRAP marker. The genome complexity of the modern hybrids comprises variable number of chromosomes between 100 and 130 [10,12,15,61,62], variable ploidy levels and copies of the homo(eo)logous chromosomes [12,17,63,64], gene duplication [17,64] and also genome modifications as insertions and deletions [12,34].

According to marker nature, is expected that genomic markers, such as AFLP and SSR markers, show higher polymorphism content than functional markers, since in transcribed regions the DNA sequences are more conserved [26,29]. However, PIC and DP averages values obtained in our study by functional TRAP markers were higher than related by other works in sugarcane [21,29,30,59,60] even when these values were compared with genomic markers [3,23]. Moreover, functional markers are more efficient for gene tagging than genomic markers and, consequently, facilitate the introgression of alleles that potentially control agronomic traits of interest by breeding programs [21,28,32,57]. Thereby, putative exclusive fragments for species or a specific accession could be evaluated through mapping association to further introgression process. Here, among the *Saccharum* genus, the *S. spontaneum* showed the highest number of putative exclusive fragments, all of them for TRAP makers related with sucrose metabolism (S5 Table). *S. spontaneum* is the wild species considered the most diverse species of the genus *Saccharum* due to its great ecogeographic distribution [65], show generally low sucrose levels and is used to introgress traits such as increased disease resistance, and ratooning [13]. So, our results suggest that *S. spontaneum* also could be promote variability for genes involved in sucrose metabolism and that these putative exclusive fragments probably have negative effects on the sucrose metabolism [23,24]. Furthermore, the low frequency of putative exclusive fragments in the BB group suggests that ancestor accessions did not encompass the

whole genetic pool used in prior breeding programs or that these new alleles observed in breeding accessions may have emerged over time as a result of changes in the genome as mutations and duplications [23].

The degree of the genetic differentiation ($\Phi_{PT}$) found in the current assignment trough AMOVA are in accordance with previous studies in sugarcane [23,66,67] and also with other polyploids crops such as sweet potato, wheat and cotton [68–70]. The population differentiation depends on the balance among migration, mutation, and drift. In polyploids species, such as sugarcane, the level of diversity within populations is naturally higher when comparing with species with lower ploidy levels [71,72]. The higher differentiation among A and BB groups suggest that there was extensive use of a small number of ancestors accessions, mainly representatives of *S. officinarum* and *S. spontaneum*, in the first interspecific crosses and also that a preferential gene complexes were fixed during breeding process to develop modern Brazilian sugarcane cultivars according to yield performance interests and environmental limitations. Furthermore, mainly in the BB group, the accessions shared a larger number of parents between them [62], which contribute to increase the divergence with the A group. On the other hand, the moderate genetic differentiation among A and FH groups and the low genetic differentiation (3%) detected among BB and FH groups suggest that FH accessions have few generations from the first breeding crossings and that may be part of the genealogy of BB accessions. Indeed, the FH accessions included in the BPSG were introduced for contribute with Brazilian breeding programs [62,74].

The results of the AMOVA are interesting, since TRAP markers were partially anchored in genes under selection process (sucrose and lignin metabolism), thus not anonymous, and even were able to detect genetic differentiation within and among accessions of the compared groups. This indicates that even for these genes there is still possibility of introgression of new alleles, opening front to germplasm exchange and assisted selection with functional molecular markers, like TRAP markers, in outcrossing heterozygous species such as sugarcane. Moreover, further studies could be conducted to determine other genes under selection with potential to differentiate populations and enable better management of crosses between and within the groups for introgression of favorable alleles [13,19,28,66,73–76].

Considering the PCA approach it was possible to verify divergences between and within the predefined groups. In the first PCA (Fig 1A), into A group, the *Erianthus* accessions were clearly divergent from the *Saccharum* accessions, supporting the taxonomic evidence which assigned each of them to a separate genus [77]. Our result agrees with other studies that used AFLP [78,79], cpSSR [80], TRAP [21,30,59], SRAP [55] and SSR [3,81] markers. The introgression of alleles of the *Erianthus* genus in sugarcane breeding programs, mainly from *E. arundinaceus*, has been evaluated in recent years to increase adaptability, disease resistance, drought resistance and biomass production [82,83]. Despite this, further studies may be conducted to evaluate other regions of the genome closely related to the outstanding traits of the *Erianthus* genus.

Thereby, when we analyzed the second PCA (Fig 1B), the close position between breeding accessions and some representatives of *S. officinarum* (for example, AJAX, CAIANA RISCADA, CAYANA, NG21-21, NG57-221, SAC OFFIC 8272 and WHITE MAURITIUS) became more evident and could be explained by the fact that this specie was one of the main ancestors of modern sugarcane cultivars, which carry 80–85% of the *S. officinarum* genetic base [15]. Furthermore, the evolutionary history of sugarcane may be inferred in the clustering of the second PCA for the A group, since the *S. barberi* accessions (AGOULE, CHIN, CHUNNE, GANDA CHENI and WHITE PARARIA) were close positioned with some *S. officinarum* accessions (CAIANA RISCADA, CAIANA VERDADEIRA, CANA BLANCA, IN84-103, NG21-17, SAC OFFIC 8272, SAC OFFIC 8276, SAC OFFIC 8280 and WHITE

MAURITIUS) and some *S. spontaneum* accessions (KRAKATAU and SES205A), possibly because *S. barberi* were originated from the hybridization of *S. officinarum* with *S. spontaneum* [17,18,84–86]. The relatedness of modern sugarcane cultivars also appears to be represented in the second PCA, since some FH accessions (NA56-79, POJ2878, TUC71-7, Co290, Co331, Co413 and Co419) used as parents in crosses to obtain Brazilian cultivars were close positioned of BB accessions, for example, the FH accession TUC71-7 was near to their progenies RB855453, RB855574 and RB855196. It is interesting to note the central position of NA56-79, which was used as parent of several accessions [74] that were located into BB subgroups in the second quadrant of the graph. On the other hand, some FH accessions, for example POJ161, Co285, Q70 and US60-31-3 were found near to accessions of the A group, suggesting that this accessions could be have few generations from the crosses between the firstly ancestors (Trop-GeneDB Sugarcane: http://tropgenedb.cirad.fr/tropgene/JSP/interface.jsp?module= SUGARCANE).

When analyzed the genetic structure through STRUCTURE software, almost all ancestors were separate of the improved accessions, especially BB accessions (Fig 2). However, the STRUCTURE results should be viewed with caution, since it is based on the assumption that all loci are considered to be in Hardy-Weinberg equilibrium within each population, without any linkage disequilibrium among loci, if they are not closely linked [87]. Thus, for complex genomes such as sugarcane, these assumptions are not fulfilled, even more when are used non-neutral markers related with traits under selection during generations [24,86,88]. Nevertheless, the comparison between PCA, STRUCTURE results and also phylogenetic tree (Fig 2), showed a good way to infer the genetic structure for BPSG.

In the phylogenetic tree, which was obtained from genetic dissimilarity matrix, there was a great similarity with the clustering seen in PCA and the almost all ancestors were grouped within a cluster such as suggested by STRUCTURE analysis. In general, the family relatedness between the BPSG accessions was present in the clusters within the clades of the phylogenetic tree. In addition, the high dissimilarity value (0.62) was found between accessions representatives of *S. officinarum* (CAIANA FITA) and *S. spontaneum* (SES205A), two morphologically distinct species used in the firstly interspecific crosses of sugarcane, while the low dissimilarity value (0.10) was found between two BB accessions, CB40-13 and RB721012. Both have in their genealogies four generations and sharing at least three ancestors, since RB721012 was obtained from a polycross (RIDESA: www.ridesaufscar.com.br; TropGeneDB Sugarcane: http:// tropgenedb.cirad.fr/tropgene/JSP/interface.jsp?module=SUGARCANE).

Furthermore, as expected, the highest average dissimilarities (0.34) were found when A group was compared with FH and BB groups. Otherwise, the lower average dissimilarity within the BB group (0.25) suggests that the Brazilian accessions shared approximately 75% of the genic regions assessed with TRAP markers and, therefore, a level of genetic uniformity for these loci between BB accessions. Similar results were found by Alwala et al. [21], Devarumath et al. [59] and Manechini et al. [15]. As a first approach to overcome this finding and considering that small number of initial parents contributed to modern hybrids [3], the incorporation of distinct genetic background may be useful to raise the genetic gain rate for the traits of interest, especially for those under high selection pressure. Despite this, although less frequently, some half-siblings (for example, CB40-13 and CB40-77) and full-sibs (for example, RB855589 and RB855598) were allocated to distinct clades, which is not uncommon in outcrossing heterozygous species, since they are characterized by high ploidy and may present genetic differences due to chromosomal inconsistencies during meiosis [10,17,63]. In this way, we can infer that the sugarcane genetic base did not narrow as much as some studies point out [20,32,89], since the genetic complexity mentioned above is able to promote variability even at loci that were possibly fixed by selection over decades. The results provide by AMOVA also corroborate

with these findings. The high linkage disequilibrium extend detected in sugarcane [24,86] regulates the exclusive allelic reservoir of each genotype that is transmitted to its progeny, which allowed the action of classic breeding programs to the present day. The use of molecular tools, as demonstrated in this study, can contribute to estimate genetic diversity and detected population structure in core collections, to increase the assertiveness of the crosses and efficiency of introgression of favorable alleles.

## Supporting information

**S1 Table. Brazilian Panel of Sugarcane Genotypes (BPSG): Accessions, pedigree information, origin and predefined groups of the 254 accessions.**
(DOCX)

**S2 Table. Names, sequences 5'– 3', genbank ID and the references of the fixed and arbitrary primers that compose TRAP markers.**
(DOCX)

**S3 Table. Functional description of the sequences that gave rise to fixed primers of TRAP markers used in this study.**
(DOCX)

**S4 Table. TRAP genotyping information.** Total number of fragments, number of polymorphic fragments, percentage of polymorphism, polymorphism information content (PIC) value and discriminatory power (DP) value for each of the eight TRAP markers evaluated in the Brazilian Panel of Sugarcane Genotypes (BPSG).
(DOCX)

**S5 Table. Putative exclusive TRAP fragments observed in the Brazilian Panel of Sugarcane Genotypes (BPSG).**
(DOCX)

**S1 Fig. Best k analysis showing k values from 2 to 9 (10 suppressed).**
(TIF)

**S2 Fig. Bootstrap analysis of TRAP genotyping.** Boxplots of the coefficients of variation (CV %), associated with the estimates of genetic dissimilarities, by bootstrap analysis for subsamples with different numbers of TRAP fragments.
(TIF)

## Author Contributions

**Conceptualization:** Monalisa Sampaio Carneiro.

**Data curation:** Monalisa Sampaio Carneiro.

**Formal analysis:** Carolina Medeiros, Thiago Willian Almeida Balsalobre, Monalisa Sampaio Carneiro.

**Funding acquisition:** Monalisa Sampaio Carneiro.

**Investigation:** Carolina Medeiros, Monalisa Sampaio Carneiro.

**Methodology:** Carolina Medeiros, Thiago Willian Almeida Balsalobre, Monalisa Sampaio Carneiro.

**Project administration:** Monalisa Sampaio Carneiro.

**Supervision:** Monalisa Sampaio Carneiro.

**Writing – original draft:** Carolina Medeiros, Thiago Willian Almeida Balsalobre.

**Writing – review & editing:** Carolina Medeiros, Thiago Willian Almeida Balsalobre, Monalisa Sampaio Carneiro.

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
