## [Decision Letter · Decision Letter 0]

14 Apr 2020

PONE-D-20-08692

Molecular diversity and genetic structure of Saccharum complex accessions

PLOS ONE

Dear Dr. Carneiro,

Thank you for submitting your manuscript to PLOS ONE. After careful consideration, we feel that it has merit but does not fully meet PLOS ONE’s publication criteria as it currently stands. Therefore, we invite you to submit a revised version of the manuscript that addresses the points raised during the review process.

We would appreciate receiving your revised manuscript by May 29 2020 11:59PM. To enhance the reproducibility of your results, we recommend that if applicable you deposit your laboratory protocols in protocols.io, where a protocol can be assigned its own identifier (DOI) such that it can be cited independently in the future. For instructions see: http://journals.plos.org/plosone/s/submission-guidelines#loc-laboratory-protocols

We look forward to receiving your revised manuscript.

Kind regards,

Tzen-Yuh Chiang

Academic Editor

PLOS ONE

Journal Requirements:

Reviewers' comments:

Reviewer's Responses to Questions

**Comments to the Author**

1. Is the manuscript technically sound, and do the data support the conclusions?

Reviewer #1: Yes

2. Has the statistical analysis been performed appropriately and rigorously? 

Reviewer #1: Yes

3. Have the authors made all data underlying the findings in their manuscript fully available?

Reviewer #1: Yes

4. Is the manuscript presented in an intelligible fashion and written in standard English?

Reviewer #1: Yes

5. Review Comments to the Author

Reviewer #1: The manuscript submitted to the Journal entitled “Molecular diversity and genetic structure of Saccharum complex accessions" is an informative piece of work and seems useful in view of harnessing the potential of this important grass species in breeding and genetics. The work will help the breeders and researcher to raise the information for identification of ideal planting material. The experimentation is properly designed and literature review is up-to-date and reflects the scope of the paper as per scope of Plos one. The conclusions are based on solid data analysis, but still few section in the manuscript is needed revision for clarity and better understanding among the readers. Based on my observations I recommend the manuscripts for minor revision.

6. PLOS authors have the option to publish the peer review history of their article (what does this mean?). If published, this will include your full peer review and any attached files.

Reviewer #1: Yes: Ram Baran Singh

---

## [Author Response · Author response to Decision Letter 0]

28 Apr 2020

Dear Editor, 

We would like to thank the important contributions of the reviewers for improving the quality of the manuscript. All suggestions/comments were duly accepted. 

Yours sincerely, 

Profa. Monalisa Sampaio Carneiro 

Plant Biotechnology Laboratory 

Department of Vegetal Biotechnology and Production 

Center for Agricultural Sciences (CCA) - Sao Carlos Federal University (UFSCAR) – SP - Brazil

---

## [Decision Letter · Decision Letter 1]

1 May 2020

Molecular diversity and genetic structure of Saccharum complex accessions

PONE-D-20-08692R1

Dear Dr. Carneiro,

We are pleased to inform you that your manuscript has been judged scientifically suitable for publication and will be formally accepted for publication once it complies with all outstanding technical requirements.

With kind regards,

Tzen-Yuh Chiang

Academic Editor

PLOS ONE

Additional Editor Comments (optional):

Reviewers' comments:

Reviewer's Responses to Questions

**Comments to the Author**

1. If the authors have adequately addressed your comments raised in a previous round of review and you feel that this manuscript is now acceptable for publication, you may indicate that here to bypass the “Comments to the Author” section, enter your conflict of interest statement in the “Confidential to Editor” section, and submit your "Accept" recommendation.

Reviewer #1: All comments have been addressed

2. Is the manuscript technically sound, and do the data support the conclusions?

Reviewer #1: Yes

3. Has the statistical analysis been performed appropriately and rigorously? 

Reviewer #1: Yes

4. Have the authors made all data underlying the findings in their manuscript fully available?

Reviewer #1: Yes

5. Is the manuscript presented in an intelligible fashion and written in standard English?

Reviewer #1: Yes

6. Review Comments to the Author

Reviewer #1: Authors did a great job to configure this research article using modern tools and technologies. It is an informative piece of work and seems useful in view of harnessing the potential of this important grass species in breeding and genetics. Based on my observations, I recommend the manuscripts for accept for publication in PLOS One.

7. PLOS authors have the option to publish the peer review history of their article (what does this mean?). If published, this will include your full peer review and any attached files.

Reviewer #1: Yes: Ram Baran Singh, International Crops Research Institute for the Semi-Arid Tropics (ICRISAT), Hyderabad, (Telangana State), Pin Code 502 324, India

---

## [Editor Report · Acceptance letter]

7 May 2020

PONE-D-20-08692R1 

Molecular diversity and genetic structure of *Saccharum* complex accessions 

Dear Dr. Carneiro:

I am pleased to inform you that your manuscript has been deemed suitable for publication in PLOS ONE. Congratulations! Your manuscript is now with our production department. 

With kind regards,

on behalf of

Dr. Tzen-Yuh Chiang 

Academic Editor

PLOS ONE